# Mechanisms of Endothelial Cell Membrane Repair: Progress and Perspectives

**DOI:** 10.3390/cells12222648

**Published:** 2023-11-17

**Authors:** Duoduo Zha, Shizhen Wang, Paula Monaghan-Nichols, Yisong Qian, Venkatesh Sampath, Mingui Fu

**Affiliations:** 1Department of Biomedical Science, School of Medicine, University of Missouri Kansas City, 2411 Holmes Street, Kansas City, MO 64108, USA; dzmkt@umkc.edu (D.Z.); nicholsap@umkc.edu (P.M.-N.); 2The National Engineering Research Center for Bioengineering Drugs and Technologies, Institute of Translational Medicine, Nanchang University, 1299 Xuefu Rd, Honggu District, Nanchang 330031, China; qianyisong@ncu.edu.cn; 3Division of Biological and Biomedical Systems, School of Science and Engineering, University of Missouri Kansas City, 5009 Rockhill Road, Kansas City, MO 64110, USA; wangshizhen@umkc.edu; 4Department of Pediatric, Children’s Mercy Hospital, Children’s Mercy Research Institute, Kansas City, MO 64108, USA; vsampath@cmh.edu

**Keywords:** endothelial cells, cell membrane repair, dysferlin, annexins, cardiovascular diseases

## Abstract

Endothelial cells are the crucial inner lining of blood vessels, which are pivotal in vascular homeostasis and integrity. However, these cells are perpetually subjected to a myriad of mechanical, chemical, and biological stresses that can compromise their plasma membranes. A sophisticated repair system involving key molecules, such as calcium, annexins, dysferlin, and MG53, is essential for maintaining endothelial viability. These components orchestrate complex mechanisms, including exocytosis and endocytosis, to repair membrane disruptions. Dysfunctions in this repair machinery, often exacerbated by aging, are linked to endothelial cell death, subsequently contributing to the onset of atherosclerosis and the progression of cardiovascular diseases (CVD) and stroke, major causes of mortality in the United States. Thus, identifying the core machinery for endothelial cell membrane repair is critically important for understanding the pathogenesis of CVD and stroke and developing novel therapeutic strategies for combating CVD and stroke. This review summarizes the recent advances in understanding the mechanisms of endothelial cell membrane repair. The future directions of this research area are also highlighted.

## 1. Introduction

Endothelial cells (ECs) form the inner lining of blood vessels that play an important role in maintaining vascular integrity and homeostasis. They provide a physical barrier between the blood and vessel walls or tissues, which is essential in regulating homeostasis and coordinating responses to blood vessel injury and inflammation. As ECs are constantly exposed to mechanical, chemical, and biological stresses, their plasma membrane is frequently disrupted and repaired. Failure to repair the disrupted membrane may cause endothelial cell death, which impairs the barrier function of blood–tissues or vessel walls, leading to vasculopathy, acute organ injury, and the initiation of atherosclerosis [1]. The capacity and mechanisms of cells to repair wounded plasma membranes have been studied in a number of cell types, especially skeletal muscle and cardiomyocytes [2,3,4,5]. Although the exact mechanisms underlying the membrane repair or resealing in these cells are still not fully understood, shreds of evidence available so far indicate that plasma membrane resealing is an autonomous and intrinsic process. In addition, although frequent disruption and resealing are commonly observed in mechanically exposed tissues, including skeletal muscle and the endothelium, the study of endothelial cell membrane repair is limited. In this review, we will discuss the recent advances in the study of endothelial cell membrane repair by focusing on the protein machinery that specifically operates in ECs. The future research direction in this area will also be highlighted.

## 2. A General Model of Cell Membrane Repair

The plasma membrane of eukaryotic cells is vulnerable to mechanical, chemical, and biological stresses. Consequently, plasma membrane disruption is a common type of cellular injury in eukaryotic cells, and effective membrane repair mechanisms have evolved to rapidly reseal a membrane breach, restore internal homeostasis, and prevent cell death [6,7]. Although the exact mechanisms underlying the membrane repair or resealing are still not fully understood, and different cell types may have different mechanisms, a general model of cell membrane repair has been hypothesized.

### 2.1. Ca^2+^ Influx as a Trigger

By actively pumping Ca^2+^ ions from the cytosol into the extracellular spaces, cells maintain cytosolic resting Ca^2+^ concentration in the nanomolar range and extracellular Ca^2+^ in the millimolar range. When the integrity of the plasma membrane is breached, a rapid influx of Ca^2+^ generates localized and transient increases in the cytosolic free Ca^2+^ concentration. The Ca^2+^ influx is required to trigger a series of reactions that rapidly reseal the disrupted membrane [8]. Early studies in sea urchin eggs showed that a Ca^2+^ influx through plasma membrane wounds triggered a rapid surface reaction followed by fully restoring the egg’s integrity, allowing subsequent fertilization and normal development. In contrast, when Ca^2+^ was removed from seawater surrounding the eggs, wounding resulted in progressive loss of cytosol and cell death [9]. These initial observations were confirmed in mammalian cells. For example, the microinjection of cells in Ca^2+^-free media inevitably results in cell death, suggesting that extracellular Ca^2+^ as a trigger is required to effectively repair microneedle wounds [10].

### 2.2. Ca^2+^-Targeted Proteins Form a Temporary Diffusion Barrier (Resealing Phase)

Different types of wounds may be repaired by different mechanisms. Tiny membrane injuries (<1 nm), such as those created by electroporation or proteins that induce a lipid disorder, may be repaired spontaneously by the force of the underlying membrane cytoskeleton [4]; however, this will not be discussed in this article. If a large membrane injury is created, a Ca^2+^ influx will rapidly activate several cytosolic Ca^2+^-binding proteins, such as calpain, dysferlin, annexins, S100A11, and transglutaminases. Calpain is a Ca^2+^-dependent cysteine protease that can rapidly cleave dysferlin to release a 72-KD C-terminal fragment (mini-DyaferlinC72). The mini-dysferlin can move to injured membrane sites and bind to the exposed lipids [11]. MG53 (also known as TRIM72) is an E3 ligase. MG53 is not a Ca^2+^ binding protein, but it can rapidly move to injured membrane sites and bind to exposed membrane lipids. MG53 can interact with dysferlin and form a tight lattice network to form a protein scaffold around the wounds, which forms the first protein barrier to reseal the wound [12,13,14]. Annexins are Ca^2+^- and phospholipid-binding proteins that move to the injured membrane sites and form a protein scaffold [15]. Annexin A2 can bind to S100A11, another Ca^2+^-activated protein, both of which are required for efficient cell membrane repair [16]. The protein complex of annexins and S00A11 may form a second protein barrier to reseal the wound. Ca^2+^-activated transglutaminase has protein-linking activity, which can make the proteins cross-link and form more intensive protein barriers at the wound sites, preventing cytosol loss while the integrity of the lipid bilayer is restored [17]. Interestingly, dysferlin can also interact with annexins to make these two layers of protein “patch” more tightly [18]. There may be more proteins involved in the formation of the protein barrier. In addition, the protein machinery resealing the wounds may be different in different cell types. For example, MG53 is a skeletal muscle-specific protein. In non-skeletal muscle cells (e.g., endothelial cells), there may be another protein or proteins serving an equivalent role as MG53.

### 2.3. Ca^2+^-Regulated Lysosome Exocytosis (Repairing Phase)

While the membrane injury is rapidly resealed, the loss of the membrane at the sites of injury needs to be repaired. Experiments in sea urchin eggs and mammalian cells revealed extensive and localized fusion of intracellular vesicles with the plasma membrane a few seconds after injury and a Ca^2+^ influx [19]. Inhibiting vesicle exocytosis by interfering with the formation of SNARE (soluble *N*-ethylmaleimide-sensitive factor attachment protein receptor) complexes impaired plasma membrane repair, suggesting that upon Ca^2+^ influx through the wound, pre-existing intracellular vesicles would fuse with each other to form a membrane “patch”, which would then merge with the injured plasma membrane and restore its integrity [20]. The identity of intracellular vesicles for membrane repair was confirmed as lysosomes [21]. Exocytosis of lysosomes has been extensively observed in injured cells, and plasma membrane repair is impaired after the inhibition of lysosome exocytosis. Impaired plasma membrane repair is also seen after the inhibition of synaptotagmin 7, a member of the synaptotagmin family of Ca^2+^ sensors present on the membrane of lysosomes [22], or of the components of SNARE complexes that mediate lysosome exocytosis [20]. Interestingly, inhibition of the lysosomal Ca^2+^ channel mucolipin-1 also impairs plasma membrane repair, suggesting that Ca^2+^ stored in the lumen of lysosomes may also contribute to the increase in the cytosolic Ca^2+^ concentration that is critical for plasma membrane resealing [23]. Recent studies suggest that other vesicles, such as endosomes and storage granules, may also be involved in the plasma membrane repair [24,25].

### 2.4. Lesion Removal by Endocytosis

The final step is the removal of the repaired lesions of the cell membrane. Mammalian cells injured mechanically or by bacterial pore-forming toxins, such as streptolysin O (SLO), were found to undergo massive endocytosis after Ca^2+^-triggered exocytosis of the lysosomes [26]. This unusual form of endocytosis observed after a Ca^2+^ influx is independent of classical endocytosis proteins, such as clathrins, requires the presence of cholesterol in the plasma membrane, and can be triggered by extracellular exposure to the enzyme sphingomyelinase, which provides an important link between lysosomal exocytosis and endocytosis-mediated plasma membrane repair. Caveolae are plasma membrane invaginations with a diameter of ~80 nm that are associated with membrane microdomains enriched in cholesterol and sphingomyelin (frequently described as lipid rafts). Caveolae are present in many cell types and are particularly abundant in cells that are under significant mechanical stress in vivo, such as muscle fibers, cardiomyocytes, and endothelial cells [27]. Interestingly, electron microscopy studies have revealed a large number of internalized vesicles with the size and morphology of caveolae in the cytoplasm of injured cells or in intact cells treated with sphingomyelinase [26]. SLO was directly visualized traveling into cells inside internalized caveolar vesicles, and the inhibition of caveolae formation blocked plasma membrane repair, strongly suggesting that wounds can be removed from the plasma membrane through caveolar endocytosis [27].

Collectively, these findings led to a multi-step model of plasma membrane repair. According to this model, a plasma membrane injury triggers a Ca^2+^ influx, the formation of a protein barrier, and exocytosis of lysosomal hydrolases, including acid sphingomyelinase (ASM), which, in turn, remodels the outer leaflet of the plasma membrane. This triggers a ceramide-dependent wave of endocytosis that internalizes the lesions (Figure 1).

## 3. Endothelial Cell Membrane Damage

The ECs in vivo are constantly exposed to mechanical, chemical, and biological stresses generated by hemodynamic forces, toxic environmental substances, and metabolites and pathogen-derived pore-forming toxins under physiological and pathological conditions. Those extracellular and intercellular factors can constantly damage endothelial cell membrane. Earlier studies found that on average, 6.5% of the endothelial cells of a rat aorta show transient and resealable plasma membrane disruption [1]. Interestingly, these wounded endothelial cells were particularly abundant around vascular bifurcation characterized by turbulent and disturbed flow, which is the area prone to atherosclerosis. These results suggest that the disruption and resealing of the endothelial plasma membrane may frequently happen, and failure to repair the wounded cells may be an important mechanism leading to the formation of early atherosclerotic lesions [1].

### 3.1. Mechanical Stresses

ECs are constantly subjected to a great amount of dynamic changed mechanical forces, including shear stress and mechanical stretch, under physiological or pathological conditions. Those mechanical stresses can directly damage endothelial cell membranes and induce cell death. For instance, under hypertension, pressure-induced stretching of the vascular wall may cause EC membrane damage, similar to exercise-induced skeletal muscle membrane damage. Other mechanical forces may also cause EC membrane damage. For example, the EC plasma membrane may be damaged by interaction with flashed blood cells or inner substances (protein aggregates or inhaled nanoparticles) [28].

### 3.2. Chemical Stresses

Oxidation of low-density lipoprotein (oxLDL) is believed to play a key role in the pathogenesis of atherosclerosis. It was demonstrated that a high dose of oxLDL can directly damage the cell membrane and cause cell death [29]. Oxidative stress and intracellular iron promote lipid peroxidation of poly-unsaturated fatty acids, leading to the removal of damaged fragments and destabilization of the plasma membrane [30]. In addition, lipid peroxidation can be induced by reactive oxygen species (ROS). During hypoxia or ischemia/reperfusion (I/R) injury, an inadequate blood supply leads to mitochondrial dysfunction and the generation of ROS, which is further exacerbated upon re-oxygenation [28]. In addition, membrane lipids are subject to enzymatic damage by host or foreign phospholipases. Alterations in membrane fluidity through interactions with amphipathic molecules or cholesterol extraction can make membranes easy to injure [28].

### 3.3. Biological Stresses

Pathogen infection can induce plasma membrane damage by several mechanisms. For example, some bacteria can breach the plasma membrane and enter the cells. To breach the plasma membrane, the bacteria can secrete virulence factors, such as secreted sphingomyelinases and phospholipases [31]. In addition, the membrane damage can also be achieved by pore-forming toxins, the largest class of bacterial toxins [32]. Although distinct from bacterial pore-forming toxins, viruses similarly compromise host membrane integrity through transmembrane channels termed viroporins [33]. There is also a large group of endogenous pore-forming proteins that can cause cell membrane damage. Perforin and complement factors are a major weapon of immune killing by pore-forming activities [34]. Other pore-forming proteins, such as gasdermins and MLKL (mixed-lineage kinase domain-like pseudokinase), which are end products of pyroptosis and necroptosis pathways, respectively, cause cell lysis by forming pores at the cell membrane [35,36].

## 4. Endothelial Cell Membrane Repair

Despite the fundamental importance of membrane repair for cell survival and tissue integrity, studies on endothelial cell membrane repair are limited. Current studies suggest that endothelial cell membrane repair may share some general mechanisms with other cell types, such as skeletal muscle, but also have specific components to fulfill the specific situations that endothelial cells encounter.

### 4.1. Dysferlin and Myoferlin

All known mammalian ferlin gene products (dysferlin, myoferlin, and otoferlin) are essential for the trafficking of intracellular vesicles and may be involved in cell membrane repair. Both dysferlin and myoferlin are abundantly expressed in skeletal muscles and are important components of repairing machinery for muscles [37]. Loss of dysferlin or myoferlin activity causes limb–girdle muscular dystrophy through impaired skeletal muscle repair after physical injury [38,39]. Otoferlin is a specific member strongly expressed in the auditory inner hair cells that are responsible for transmitting auditory information to the central nervous system. Mutations of otoferlin cause autosomal recessive deafness in humans, and genetic inactivation of the otoferlin gene impairs Ca^2+^-dependent exocytosis in mouse auditory inner hair cells and causes profound deafness [40]. Proteomic profiling showed that both dysferlin and myoferlin are localized at the endothelial membrane and lipid rafts [41]. Western blot confirmed their expression in human primary endothelial cells. Moreover, recent studies show that dysferlin may participate in endothelial cell membrane repair by mediating lysosome fusion with the endothelial cell membrane [42], and myoferlin is required for efficient endothelial membrane repair after mechanical injury [43]. Nevertheless, the importance and mechanisms of dysferlin and myoferlin in endothelial cell membrane repair in vitro and in vivo need to be further determined.

### 4.2. Annexins and S100A10/11

Annexins constitute a multigene family of Ca^2+^-dependent phospholipid-binding proteins, with members of the family being expressed in animals, plants, fungi, and protists. This family can be divided into five classes (A–E) based on their biological origins. In vertebrates, class A is composed of 12 members, AnxA1–AnxA13 (Anx12 is currently unassigned). Annexins have two structural domains: a conserved C-terminal domain with 4 repeats of 70–80 amino acids (8 in the case of AnxA6) and a variable N-terminal domain. Each repeat contains a Ca^2+^ binding motif, allowing the annexins to rapidly translocate to the plasma membrane or intracellular membranes by binding to negatively charged phospholipids [44]. The functions of annexins include Ca^2+^-regulated endocytic and exocytic events, the maintenance and regulation of membrane–cytoskeleton contacts, and membrane domain organization [45]. AnxA2 is the most abundant member in endothelial cells [46]. A1 and A5 are also expressed in endothelial cells. Recent studies show that AnxA1, A2, A5, and A6 all rapidly move to the wounded sites after injury in endothelial cells. They can form a protein complex with S100A11, which is an EF hand-type Ca^2+^-binding protein. The knocking down of AnxA2 or S100A11 proved that both of them are required for efficient repair of endothelial cell membrane wounds. Further studies show that the S100A11 C-terminal extension (residues 93–105) can interact with AnxA1 and AnxA2 in a Ca^2+^-dependent manner. In addition, synaptotagmin-1 also binds to S100A11 in the presence of Ca^2+^, and depletion of synaptotagmin-1 interferes with wound site recruitment of S100A11 and proper membrane resealing in endothelial cells [47].

### 4.3. Caveolins

Caveolins are major components associated with caveolar lipid rafts in the plasma membrane and are implicated in many aspects of cellular functions. There are three members of caveolins: caveolin-1, caveolin-2, and caveolin-3. Caveolin-1 is the primary member of caveolins and is ubiquitously expressed in all tissues, including endothelial cells. Caveolin-3 (Cav3) is strictly expressed in skeletal muscles. Mutation of Caveolin-3 is linked to muscular dystrophy in patients [28]. Cav3 is essential for the repair of muscle membrane damage. Cav3 can form a protein complex with MG53 and dysferlin. Cav3 mutants that cause retention of Cav3 in the Golgi apparatus result in aberrant localization of MG53 and dysferlin in a dominant-negative fashion, leading to defective membrane repair [48]. These results suggest that Cav3 at the membrane is required for correct localization and retention of MG53 and dysferlin on muscle membranes. While Cav3 is specifically expressed in muscles, Cav1 is expressed in many tissues, including endothelial cells. Cav1 forms a protein complex with myoferlin in endothelial cells. Both Cav1 and myoferlin are critical for efficient endothelial membrane repair after injury [43].

### 4.4. Weibel–Palade Bodies

Weibel–Palade bodies (WPB) are unique secretory organelles in endothelial cells that undergo evoked exocytosis following intracellular Ca^2+^ or cAMP elevation, thereby supplying the vasculature with factors controlling hemostasis. The major components stored within WPB are the multimeric glycoprotein von Willebrand factor (VWF) and the adhesion receptor P-selectin. WPBs move to the endothelial cell membrane when there is a blood vessel injury and fuse with the plasma membrane. VWF is released into the lumen of the blood vessel and provides platelet adhesion sites. Low levels of VWF can lead to a bleeding tendency, such as in von Willebrand disease, and elevated levels of VWF are associated with an increased risk for thrombosis and cardiovascular disease [49]. A recent study observed that complement-mediated endothelial membrane injury can be mitigated by the mobilization of WPBs along with the secretion of VWF. Endothelial cells lacking WPBs were not resistant to complement-mediated damage but became resistant when transfected to express VWF. The study suggests that WPB exocytosis in response to endothelial plasma membrane damage may be an essential mechanism for endothelial cell membrane repair [50]. Interestingly, several other studies show that annexin A2 and S100A10 complex is required for WPB exocytosis [51,52]. In a recent study, the Annexin A2/S100A10 was confirmed to be required for efficient endothelial cell membrane repair after mechanical injury [46]. How WPB repairs the endothelial cell membrane still needs further study.

### 4.5. Other Proteins

Theoretically, any proteins that regulate processes such as cytoskeleton, exocytosis, and endocytosis may affect cell membrane repair. For example, Rab3a, Rab10, myosin 11A, 11B, SNARE, and ESCRT III (endosomal sorting complexes required for transport III) are reported to be involved in the regulation of cell membrane repair [53,54,55,56]. However, whether they are also involved in endothelial cell membrane repair needs to be determined. Anoctamin 5 (ANO5, also known as TMEM16E) is a member of the transmembrane 16 (TMEM16) family of Ca^2+^-activated ion channels and phospholipid scramblases. Human mutations of ANO5 cause adult-onset limb–girdle muscular dystrophy 2L (LGMD2L) and Miyoshi muscular dystrophy (MMD3). In mice, ANO5 is implicated in muscle differentiation and membrane repair [57]. ANO5 is primarily expressed in muscle and bone. It is also expressed in endothelial cells; however, its role in endothelial cells remains unclear. Endophylin-A2, which participates in membrane vesiculation during receptor-mediated endocytosis, is a ~40 kDa SH3 domain-containing protein that binds to the proline/arginine-rich domain of dynamin 2, a ~100 kDa GTPase that is essential for endocytic membrane scission. Both endophilin-A2 and dynamin 2 are involved in the regulation of cell membrane repair [58]; however, whether those proteins are also involved in endothelial cell membrane repair remains to be studied. MG53 (also known as TRIM72) is a protein member of the TRIM family that acts as a key component of cell membrane repair machinery in muscles [14]; however, MG53 is not expressed in human endothelial cells. The role of MG53 may be fulfilled by another TRIM protein or non-TRIM protein in endothelial cells, but this needs to be determined.

## 5. The Diseases Linked with Defective Endothelial Membrane Repair

Failure to rapidly reseal a damaged endothelial membrane may result in endothelial cell death, which is linked with acute or chronic vascular diseases, such as disruption of the blood–tissue barrier or atherosclerosis.

### 5.1. Atherosclerosis

Atherosclerosis is a common condition that links to cardiovascular diseases and stroke and is associated with local inflammation and oxidative stress. Disruption of endothelial cell homeostasis is an early step in the atherosclerosis process, as it can lead to vascular permeability, impaired clotting function, and lipid infiltration into perivascular intima, which causes stress injuries to the vascular wall. It was established that dysferlin, myoferlin, annexins, and caveolins are actively involved in the process of endothelial membrane repair and other functions in endothelial cells. It may be expected that the loss of function of those proteins may promote the development of atherosclerosis. Despite high dysferlin expression in mouse and human atheromatous plaques, the loss of dysferlin did not affect the atherosclerotic burden as measured in the aortic root, arch, and thoracic and abdominal aortic regions in ApoE−/− mice [59]. These results may be due to myoferlin compensating for the role of dysferlin in endothelial cells. In addition, genetic ablation of caveolin-1, a principal structural protein component of caveolae, which are small invaginations in the plasma membrane, was proven to protect against the development of atherosclerosis, with about a 65% reduction in the atherosclerotic lesion area [60]. Annexin A1 is a 37 kDa protein that plays an important role in the cardiovascular system. Annexin A1 is found in different locations, including the plasma membrane, endosomes, secretory vesicles, cytoskeleton, and nucleus. It can also be secreted outside of cells when the cells are activated. Several studies have demonstrated that annexin A1 protects from atherogenesis and atheroprogression in mice [61]. Annexin A2 expression is markedly upregulated in atherosclerotic lesions of ApoE knockout mice [62]. However, germline deletions of Annexin A2 did not reduce the atherosclerotic burden in ApoE-deficient mice [63].

### 5.2. Diabetes Mellitus

Diabetes mellitus is a metabolic disorder characterized by hyperglycemia due to insufficient insulin production or insulin resistance. Vascular injury arising from hyperglycemia is considered a major contributor to the growing morbidity and mortality rates associated with diabetes. In diabetic patients, oxidative stress, reduced nitric oxide (NO) levels, and inflammation all can contribute to the disruption of endothelial integrity, leading to abnormal blood fluidity and vascular tone. Instant cell membrane resealing (ICMR) is a crucial mechanism that helps to maintain endothelial integrity. Annexin A5 can form a two-dimensional bandage at the torn membrane edges to prevent the expansion of the membrane wound and promote the final step of membrane resealing. However, more retarded ICMR was observed in mouse aortic endothelial cells exposed to high glucose, which is because high glucose decreased the association of membrane ceramide with annexin A5 [64]. The high glucose-induced impairment of membrane resealing could be prevented by sphingomyelin or C24-ceramide pretreatment [65]. Additionally, hyperglycemia also elevates lysosome exocytosis, ceramide production, and membrane raft clustering. Overall, high glucose impairs endothelial membrane resealing and makes the cells more vulnerable to secondary membrane damage.

### 5.3. Acute Respiratory Distress Syndrome (ARDS) and Idiopathic Pulmonary Fibrosis (IPF)

Cell injury and death play a critical role in lung-injurious diseases. In ARDS and IPF, the lung is exposed to various insults, such as infection, oxidative stress, and mechanical stress, and is highly susceptible to plasma membrane wounding. Cell death always occurs through apoptosis, necrosis, necroptosis, and other undefined mechanisms following plasma membrane wounding. Lung injury and repair are involved in the pathogenesis of ARDS and IPF. Prolonged cell wounding and aberrant repair make it difficult to maintain cell membrane integrity, resulting in the release of intracellular contents and a subsequent immune response. Proinflammatory cytokines can also activate downstream signaling, which either produces membrane injury productions or direct compromised plasma membrane integrity as part of the cell death mechanism. Quick and effective plasma membrane repair can restore the injurious cells and preserve cellular homeostasis. In type I alveolar epithelial cells, MG53 improves cellular integrity through its interaction with caveolin 1 [66]; however, the link between MG53 and caveolar endocytosis or other protein-binding partners at the plasma membrane in the context of repairing the wounding of lung endothelial cells is not well understood.

## 6. The Resealing Agents for Endothelial Membrane Repair

When the endothelial cell membrane is damaged, it may lead to cell death and have various adverse consequences; therefore, treatment targeting membrane repair may help to maintain the integrity of endothelial cells and benefit patients with membrane repair defects.

### 6.1. Recombinant MG53

MG53, also known as TRIM72, has been identified as an essential component of the cell membrane repair machinery in striated muscles. The recombinant human MG53 can specifically be transported to the membrane wounding sites and prevent the expansion of the rupture. Several studies have demonstrated that treatment with recombinant human MG53 protein increased membrane integrity and protected against organ injury in mice, such as in the muscles, heart, lungs, kidney, brain, and liver [67,68,69,70,71,72].

### 6.2. Recombinant Annexins

Several annexins can be secreted outside of cells. The recombinant human annexins exhibit many effects on cells, including membrane repair, anti-inflammation, promoting fibrinolysis, etc. In numerous studies, treatment with recombinant human annexins and annexin analog peptides have consistently found positive outcomes in animal models of sepsis, myocardial infarction, and ischemia-reperfusion injury. Annexins A1 and A5 improve organ function and reduce mortality in animal sepsis models, inhibit inflammatory processes, reduce inflammatory mediator release, and protect against ischemic injury. The mechanisms of action and demonstrated efficacy of annexins in animal models support the development of annexins and their analogs for the treatment of sepsis. The safety and efficacy of recombinant human annexin A5 are currently being studied in clinical trials in septic patients [73].

### 6.3. Poloxamer 188

Poloxamer 188 (P188) is a stable amphiphilic polymer localized in lipid monolayers and damaged regions of membranes. The FDA approved it as a therapeutic reagent to reduce blood viscosity for transfusions. It has been shown to increase the structural stability and resealing of the plasma membrane, making it a potential therapeutic agent for various conditions that involve membrane damage. For example, the administration of P188 can reduce the loss of muscle mass in dysferlin-deficient SJL mice, which could provide a basis for potential therapeutic strategies for dysferlinopathy [74]. In addition, P188 can instantly improve ventricular geometry and block the development of acute cardiac failure in dystrophic mice during a dobutamine-mediated stress protocol [75]. P188 pretreatment could restore brain–blood barrier (BBB) integrity, suppress traumatic brain injury (TBI)-induced neural cell death, and improve neurological function [76]. Notably, a recent study demonstrated that P188 could prolong endothelial cell survival in bovine corneas stored at 4 °C by lowering the surface tension of the cell membranes [77]. The safety and non-toxicity of P188 in humans was confirmed, but high doses may be toxic for long-term use; therefore, more research is needed to understand the potential benefits and risks of P188.

### 6.4. Acid Sphingomyelinase (ASM), Sphingomyelin, or Ceramide

Plasma membrane repair requires Ca^2+^-dependent endocytosis to remove membrane lesions. Membrane rafts are specialized domains in the plasma membrane that are enriched in certain lipids, such as sphingomyelin and glycosphingolipids, as well as cholesterol. The clustering of membrane rafts can be mediated by the translocation of acid sphingomyelinase (ASM) to these domains. In response to certain stimuli, such as cell injury or stress, ASM can translocate to the site of injury and become activated to facilitate repair. The local ASM activation and ceramide production is an important step that mediates membrane raft clustering. Ceramide-mediated endocytosis could repair membrane wounding quickly. Exogenously added recombinant ASM restores endocytosis and membrane resealing in ASM-depleted cells [78]. Chen et al. also suggest that ceramide-mediated instant cell membrane repair in endothelial cells is impaired during diabetes, and this impairment could be prevented by sphingomyelin and ceramide pretreatment [64]. In addition, caspase-7 could enhance the endocytic membrane repair of GSDMD and MLKL pore opening by activating ASM to produce ceramide [79]. The information on resealing agents has been summarized in Table 1.

## 7. Discussion

ECs are constantly exposed to mechanical, chemical, and biological stresses generated by hemodynamic forces, toxic environmental substances and metabolites, and pathogen-derived pore-forming toxins under physiological and pathological conditions. A complex reparative/regeneration system is required to repair the wounded membrane to keep cells surviving. However, despite the fundamental importance of membrane repair for cell survival and vascular integrity, studies on endothelial cell membrane repair are limited. Current studies suggest that endothelial cell membrane repair may share some general mechanisms with other cell types, such as skeletal muscle, but also have specific components to fulfill the specific challenges/stimuli that endothelial cells encounter. The core machinery for endothelial cell membrane repair remains to be determined along with injury-specific (sepsis, shear stress, metabolic insults) repair mechanisms. Some of the proteins identified as playing a role in endothelial plasma membrane repair include dysferlin, myoferlin, annexins, and caveolins, as described above. There may be other proteins involved in this process. In the future, (1) the importance of cell membrane repair in the pathogenesis of endothelial cell-related diseases needs to be evaluated; (2) the core machinery and its working mechanisms of endothelial cell membrane repair needs to be identified; and (3) the follow-up effect of endothelial cell injury and repair on endothelial function needs to be further studied. Identifying the mechanisms might enable the future application of recombinant molecular therapy to reconstitute the EC membrane and attenuate progress of diseases.

## Figures and Tables

**Figure 1 cells-12-02648-f001:**
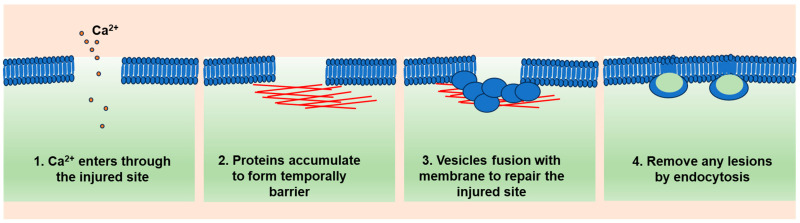
A general model of cell membrane repair. When the cell membrane is penetrated, the first event is extracellular Ca^2+^ entering the cells through the injured site (**1**). Then, some Ca^2+^-sensitive proteins, such as dysferlin and annexins, will quickly move to the injured site and form oligomers to reseal the injured site (**2**). Next, cellular vesicles will move further to the site and repair the damaged membrane (**3**). Finally, the repaired lesions will be removed by endocytosis (**4**).

**Table 1 cells-12-02648-t001:** The resealing agents for cell membrane repair.

Name	Action	Experimental Conditions	References
Reconbinant MG53	Increased membrane integrity and protects against organ injury	Mice, in vivo injection	[67,68,69,70,71,72]
Recombinant annexins	Improve the outcome of sepsis, myocardial infarction, and ischemia-reperfusion injury	Mice, in vivo injection	
		Clinical trial	[73]
Poloxamer 188	Reduce blood viscosity for transfusion	FDA-approved	
	Improve outcome of TBI	Mice, in vivo injection	[76]
	Prolong endothelial cell survival	Bovine corneas, in vitro	[77]
Acid sphingomyelinase	Promote membrane resealing	Cells, in vitro	[78]
Sphingomyelin and ceramide	Protect endothelial cell membrane impairment during diabetes	Cells, in vitro Mice, in vivo	[64]

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
