# Peer review of "Mechanisms of Endothelial Cell Membrane Repair: Progress and Perspectives"

_cells, 2023, doi:10.3390/cells12222648_

Round 1

Reviewer 1 Report

Comments and Suggestions for Authors

This is a very good review article on a novel topic of a potential clinical implications.

1) There are numerous grammar mistakes in the manuscript. For example, in line 74, “it may repair” should be “it may be repaired”  and so on.

2) The abbreviations in the manuscript should be thoroughly revised:

Line 102: SNARE is not explained

Line 120: SLO is not added to list of abbreviations placed at the end of the manuscript.

Line 138: ASM abbreviation is introduced for the first time without explanation.

Line 359: ASM abbreviation is explained again although it is explained in line 254 and it should be first explained in line 138

3) Section 5: The resealing agents for endothelial membrane repair

The findings of the studies referenced in this section can be better presented in a table to differentiate between results of in vivo, in vitro and clinical studies. More details should be added about the conditions of these studies.

Comments on the Quality of English Language

There are numerous grammar mistakes in the manuscript. 

Author Response

Response for the reviewer’s comments:

  1. This is a very good review article on a novel topic of a potential clinical implications.

Response: Thank you for your positive comment!

  1. There are numerous grammar mistakes in the manuscript. For example, in line 74, “it may repair” should be “it may be repaired” and so on.

Response: Thank you for your carefully reading! Now in the revised version, we have carefully revised the text to clean up those grammar mistakes and we have asked a native speaker to proof the whole manuscript.

  1. The abbreviations in the manuscript should be thoroughly revised:

Line 102: SNARE is not explained

Line 120: SLO is not added to list of abbreviations placed at the end of the manuscript.

Line 138: ASM abbreviation is introduced for the first time without explanation.

Line 359: ASM abbreviation is explained again although it is explained in line 254 and it should be first explained in line 138

Response: Yes, those abbreviations have been properly explained when they appear at first time.

  1. Section 5: The resealing agents for endothelial membrane repair. The findings of the studies referenced in this section can be better presented in a table to differentiate between results of in vivo, in vitro and clinical studies. More details should be added about the conditions of these studies.

Response: We have now expanded the manuscript by adding a new paragraph in page 4-5. To make the manuscript concisely, it is not necessary to repeat the information as a table.

Reviewer 2 Report

Comments and Suggestions for Authors

It is too short for a review article on the subject and it is poorly illustrated whatever the subject is potentially of a major interest if well done

Comments on the Quality of English Language

I

Author Response

Response for the reviewer’s comments:

  1. It is too short for a review article on the subject and it is poorly illustrated whatever the subject is potentially of a major interest if well done.

Response: We have now expanded the text by adding a new paragraph “Endothelial cell membrane damage” (see page 4-5). Since the subject is not well studied, there is not so much information about this subject.

Reviewer 3 Report

Comments and Suggestions for Authors

The review by Zha et al, entitled “Mechanisms of endothelial cell membrane repair: progress and perspectives” addresses an important topic regarding membrane repair systems in endothelial cells. The molecules and mechanisms involved in sophisticated membrane repair system are described in the review. However, there are some issues that need to be addressed:

- In the abstract, the authors report the importance of membrane repair in the pathogenesis of cardiovascular diseases (CVD) and stroke. They should explore this aspect further in the review.

- Since inflammation and oxidative stress play an important role in vascular damage, the authors know how inflammatory mediators and reactive oxygen species (ROS) alter the mechanisms involved in endothelial cell membrane repair.

- Authors should include a caption in Figure 1.

- There are some unspecified abbreviations, such as TBI (line 348), please enter the full name.

Author Response

Response for the reviewer’s comments:

  1. In the abstract, the authors report the importance of membrane repair in the pathogenesis of cardiovascular diseases (CVD) and stroke. They should explore this aspect further in the review.

Response: This is summarized in “5.1 Atherosclerosis”.

  1. Since inflammation and oxidative stress play an important role in vascular damage, the authors know how inflammatory mediators and reactive oxygen species (ROS) alter the mechanisms involved in endothelial cell membrane repair.

Response: We have now added a new paragraph to discuss the damages of endothelial cell membrane by these pathogenic factors in the page 4-5.

  1. Authors should include a caption in Figure 1.

Response: Yes, the legend for this figure is included.

  1. There are some unspecified abbreviations, such as TBI (line 348), please enter the full name.

Response: We have checked the whole manuscript and added descriptions for those abbreviations at their first appeared place.

ttachment.

Round 2

Reviewer 2 Report

Comments and Suggestions for Authors

The authors made little change in the text and did not improved illustration 

Comments on the Quality of English Language

The language is correct but elementary

Author Response

The Reviewer's Comment: The authors made little change in the text and did not improved illustration.

Response: We have now added a table to summarize the information on resealing reagents.